# Salt Water Exposure Exacerbates the Negative Response of *Phragmites australis* Haplotypes to Sea-Level Rise

**DOI:** 10.3390/plants13060906

**Published:** 2024-03-21

**Authors:** Austin Lynn, Tracy Elsey-Quirk

**Affiliations:** Department of Oceanography & Coastal Sciences, Louisiana State University, Baton Rouge, LA 70803, USA; tquirk@lsu.edu

**Keywords:** sea-level rise, dieback, stress, wetlands, invasion, restoration, habitat loss, ecosystem services

## Abstract

The response of coastal wetlands to sea-level rise (SLR) largely depends on the tolerance of individual plant species to inundation stress and, in brackish and freshwater wetlands, exposure to higher salinities. *Phragmites australis* is a cosmopolitan wetland reed that grows in saline to freshwater marshes. *P. australis* has many genetically distinct haplotypes, some of which are invasive and the focus of considerable research and management. However, the relative response of *P. australis* haplotypes to SLR is not well known, despite the importance of predicting future distribution changes and understanding its role in marsh response and resilience to SLR. Here, we use a marsh organ experiment to test how factors associated with sea level rise—inundation and seawater exposure—affect the porewater chemistry and growth response of three *P. australis* haplotypes along the northern Gulf of Mexico coast. We planted three *P. australis* lineages (Delta, European, and Gulf) into marsh organs at five different elevations in channels at two locations, representing a low (Mississippi River Birdsfoot delta; 0–13 ppt) and high exposure to salinity (Mermentau basin; 6–18 ppt) for two growing seasons. Haplotypes responded differently to flooding and site conditions; the Delta haplotype was more resilient to high salinity, while the Gulf type was less susceptible to flood stress in the freshwater site. Survivorship across haplotypes after two growing seasons was 42% lower at the brackish site than at the freshwater site, associated with high salinity and sulfide concentrations. Flooding greater than 19% of the time led to lower survival across both sites linked to high concentrations of acetic acid in the porewater. Increased flood duration was negatively correlated with live aboveground biomass in the high-salinity site (χ^2^ = 10.37, *p* = 0.001), while no such relationship was detected in the low-salinity site, indicating that flood tolerance is greater under freshwater conditions. These results show that the vulnerability of all haplotypes of *P. australis* to rising sea levels depends on exposure to saline water and that a combination of flooding and salinity may help control invasive haplotypes.

## 1. Introduction

Climate change can cause an increase in plant stressors through drought, sea-level rise, and biotic invasions, which, in some cases, can lead to extensive vegetation mortality and ecosystem collapse [1,2]. Coastal wetlands are particularly vulnerable to climate-related changes such as sea-level rise (SLR), saltwater intrusion, and drought [3], all of which are projected to increase in the future [4,5]. Vegetation plays a key role in coastal wetland resilience to SLR by contributing organic matter and trapping sediment, which increases surface elevation as flooding increases [6,7,8]. Thus, wetland ecosystem vulnerability to SLR largely depends on the tolerance and response of plants to inundation and salinity.

*Phragmites australis* (Cav.) Trin. Ex Steaud is a genetically diverse perennial grass species that grows in fresh and brackish marshes globally [9,10]. *P. australis* has many distinct haplotypes, which differ genetically, morphologically, and ecologically [11,12]. Some *P. australis* haplotypes are aggressive invaders such as the European haplotype, which competes with a native haplotype along the Atlantic coast and the Great Lakes of North America and reduces wetland biodiversity by displacing other native plant species [10,13]. Along the northern Gulf of Mexico coast, three of the most common *P. australis* haplotype lineages are Delta (haplotype M1), Gulf (also known as Land type, haplotype I2, or *Phragmites australis* ssp. *berlandieri* E Fourn.), and European (EU, haplotype M) [14,15]. The Delta type is the most widespread and common in low-elevation marshes while the Gulf type occurs slightly landward along channel banks and ditches. Both Delta and Gulf types are considered to be naturalized [16]. The invasive EU type is rare along the coast, occurring only in the Mississippi River ‘Birdsfoot’ Delta and in patches amongst the Delta type. This genetic diversity is relevant to the plant’s response to sea-level rise, as the different *P. australis* lineages partition niches across marsh types and exhibit varying tolerance to stressors such as salinity [17,18]. Yet, it is unknown whether these niche preferences translate to varying responses to SLR.

Wetland plants are generally adapted to flooded conditions; however, SLR can induce stress by increasing the amount of time the plants experience anoxic conditions. In particular, a lack of oxygen in marsh soils contributes to the buildup of toxic sulfides and volatile fatty acids in the porewater, which can cause a reduction in growth and mortality at high concentrations [19,20,21]. Marsh organ studies are valuable for testing how rising sea levels impact plant performance [22,23,24]. *Spartina alterniflora* Loisel. biomass exhibits a parabolic relationship with flooding, with the highest productivity at a moderate level of inundation in North Inlet, South Carolina [22,25]. However, along the northern Gulf of Mexico coast, the productivity of *S. alterniflora* declined with an increase in flooding [22], these divergent responses are likely associated with a lower tidal range on the Gulf Coast. For brackish species such as *Spartina patens* (Ait.) Muhl., flood stress compounds salinity stress such that greater flood duration more dramatically reduces plant growth under higher salt concentrations [22]. Using marsh organs to explore *P. australis* response to SLR is crucial, as *P. australis* occurs across much of the high marsh along the US Atlantic and Gulf Coasts and has been shown to play an important role in salt marsh transgression [26]. In particular, *P. australis* inhibits the landward expansion of salt marsh under low-salinity conditions, while high salinities significantly favor the growth of salt marsh grasses over that of *P. australis* [26]. While in many areas, land managers are interested in the conditions to control or eliminate its spread, *P. australis* is valued in the Mississippi River Birdsfoot Delta, where rates of relative SLR are extremely high for its ability to stabilize marshes, trap mineral sediment, and build marsh elevation through organic accretion [27,28]. However, *P. australis* in the Birdsfoot Delta has recently been experiencing dieback, similar to that documented in Europe and elsewhere [29,30], and many factors have been suggested as the cause, notably, herbivory damage and salinity pulses [31]. Thus, testing the response of *P. australis* to increasing inundation and salinity will inform the response to SLR, management of invasive haplotypes, as well as potential causes of dieback.

To investigate the effect of sea-level rise and seawater exposure on *P. australis* haplotypes, we employed a marsh organ mesocosm study at two locations along the Louisiana coastal plain. We hypothesize that (1) increased inundation time and the associated accumulation of soil phytotoxins (e.g., volatile fatty acids, sulfides) reduces *P. australis* growth and survival; (2) negative responses to flooding and phytotoxins are exacerbated in high-salinity conditions compared to low-salinity conditions; and (3) *P. australis* lineages differ in response to sea-level rise-associated abiotic stressors.

## 2. Results

### 2.1. Site Differences and Correlations across Abiotic Factors

The percent time flooded of each marsh organ elevation level differed slightly from the original targets due to annual variations in water levels. Nonetheless, the pipe elevations captured a wide range of potential flooding durations (3.5–100% in MRD and 3–100% flooded in Rockefeller; Figure 1). Water levels were lower and salinity concentrations were generally much greater in Rockefeller throughout the study duration, as expected (Figure 2).

At the end of the first growing season, acetic acid concentration was greater in treatments with more flooding at both sites (χ^2^ = 5.18, *p* = 0.02; Figure 3b). Porewater sulfide concentration was significantly higher at the end of the second growing season than after the first growing season at both sites (MRD *p* = 0.01, Rockefeller *p* = 0.003). Sulfide concentration in Year 1 was similar between sites but, in Year 2, it was significantly greater in Rockefeller (2190.61 ± 327.86 μmol/L) than MRD (555.50 ± 148.82 μmol/L; χ^2^ = 15.95, *p* < 0.0001; Figure 4). Porewater sulfide concentration was high in all but the highest elevation treatment in Rockefeller and significantly increased with greater time flooded in the MRD in Year 2 (χ^2^ = 5.64, *p* = 0.018 Figure 3). Redox potential was lower in Rockefeller than MRD (χ^2^ = 4.56, *p* = 0.033), and was negatively associated with flood duration across sites (χ^2^ = 6.64, *p* = 0.01; Figure 5).

### 2.2. Plant Survival

During the first growing season, Phragmites haplotypes significantly differed in survival percentage (EU: 68%, Delta: 74%, Gulf: 78%; χ^2^: 11.95, *p* = 0.0025). Furthermore, there was a significant interaction between the site and flooding level, such that flood stress led to much greater mortality in the high-salinity site compared to the fresh site (χ^2^: 135.76, *p* < 0.001; Figure 6a,b).

At the end of the second growing season, site and haplotype had a significant interaction in predicting survival, such that EU and Gulf haplotypes survived better in the fresh MRD site compared to Rockefeller (Chisq: 11.57, *p* = 0.0031; Figure 6c,d). During year two in the MRD, the Delta haplotype had the lowest average survivability at 46%, followed by EU at 60% and Gulf at 72%, while in Rockefeller, Delta had the highest survival at 26%, followed by EU at 16% and Gulf at 10%. Moreover, flooding duration and lineage had a significant interaction during the second growing season (χ^2^ = 6.73, *p* = 0.035). While Phragmites' survival generally declined with increased flooding, some haplotypes had a sharper decline than others (Figure 6c,d). Survivorship was greater in MRD than Rockefeller in both growing seasons (Year 1: χ^2^ = 38.15, *p* < 0.001; Year 2: χ^2^ = 25.18, *p* < 0.0001; Figure 5).

Phragmites mortality was related to high acetic acid concentration in the first growing season across both sites (χ^2^ = 12.10, *p* < 0.001; Figure 3a). Phragmites mortality was not associated with other volatile fatty acids (propionic, isobutyric, butyric, isovaleric, valeric, isocaproic, caproic, and heptanoic acid). Redox potential and sulfide concentration also were not significant predictors of mortality.

### 2.3. Aboveground Biomass

Across both sites, aboveground biomass differed between lineage depending on flooding (χ^2^ = 7.5447, *p* = 0.024), while no other interaction was significant. This is evident in that the Gulf haplotype had a more positive response to flooding than other haplotypes (Figure 7e,f). Aboveground biomass was also significantly greater in the freshwater site than in the high-salinity site (χ^2^ = 12.36, *p* < 0.0005).

Sulfide concentration (χ^2^ = 11.44, *p* = 0.0007), percent time flooding (χ^2^ = 10.37, *p* = 0.001), and the interaction between the two (χ^2^ = 10.01, *p* = 0.002) negatively affected aboveground biomass in RWR, while redox potential had no such relationship. Of the plants sampled for sulfides in Rockefeller, only two had live aboveground biomass. Of those two, the plant with 148 μmol/L of sulfide was in the 1% flooding treatment, and the plant with 2598 μmol/L was flooded 81% of the time (Figure 8). Aboveground biomass also significantly differed across plant lineages in Rockefeller: Delta produced the most (2.66g + 0.84), then Gulf type (1.07g ± 0.69), and EU produced the least (0.68g ± 0.34; χ^2^ = 9.80, *p* = 0.007; Figure 7). Aboveground biomass of Delta type was negatively related to percent time flooding (χ^2^ = 10.42, *p* < 0.01) and porewater sulfide concentration (χ^2^ = 13.16, *p* < 0.001), while aboveground biomass of Gulf and EU was not (Figure 7).

In the MRD, live aboveground biomass was greatest for Gulf type (49.58 g ± 9.11 SE), followed by EU (21.58 g ± 5.63) and Delta type (14.69 g ± 3.50; χ^2^ = 9.0882, df = 2, *p* = 0.01). Aboveground biomass was not related to the percent time flooded, porewater sulfide concentration, or redox potential in the MRD.

### 2.4. Belowground Biomass

Site and lineage had an interactive effect on belowground biomass, with the Delta haplotype producing the most biomass in the saline site Rockefeller, but the least amount in the MRD (Figure 6b; χ^2^ = 6.4643, *p* = 0.039). In the MRD, the Gulf haplotype produced the most belowground biomass (54.23 g ± 8.16), followed by EU (34.93 g ± 9.67) and Delta (26.64 g ± 4.64). In Rockefeller, Delta produced the most average belowground biomass (6.45 g + 1.76), followed by Gulf haplotype (2.54 g + 1.06), and then EU (1.49g ± 1.12). Percent time flooded was also negatively associated with belowground biomass across both sites (χ^2^ = 9.079, *p* = 0.0026).

Belowground biomass in Rockefeller was not associated with flooding or sulfides. Live belowground biomass in the MRD significantly declined with increasing porewater sulfide (χ^2^ = 4.98, *p* = 0.03, Figure 8b), and exhibited a nonsignificant trend to decline with increasing flood levels (χ^2^ = 3.68, *p* = 0.055). Soil redox potential had no influence on aboveground and belowground biomass accumulation across both sites.

## 3. Discussion

Our findings illustrate how the stress response to sea-level rise of an important wetland plant species differs dramatically depending on plant haplotype and ecological context. *P. australis* growth varied widely between saline and, such that survival and biomass accumulation were, on average, 42% and 1157% higher, respectively, in the fresh site. *P. australis* was also susceptible to flood stress, exhibiting consistently lower survivorship in heavily flooded treatments (Figure 6), while drought stress was not apparent in the high-elevation treatments, even in the saline site (but compared to drought stress found by [32]). This contrasts with a marsh organ study of an invasive, potentially more flood-tolerant freshwater species, *Colocasia esculenta* L. Schott, which found a positive relationship between aboveground biomass and flood duration [33,34]. Porewater phytotoxins such as sulfide and acetic acid, associated with salinity and anaerobic conditions from flooding, further contribute to sea-level rise-associated stress, including reduced growth and mortality in *P. australis* (Figure 3 and Figure 8; Refs. [35,36]). Ultimately, such dieback conditions in fresh and brackish marshes may be repopulated by salt marsh species if propagule sources are nearby [37]. Our work contributes to a growing body of research demonstrating how coastal freshwater and brackish wetland plant stress will be exacerbated by projected future sea-level rise, and the combined effects of increased flooding and salinity [38,39].

### 3.1. Phragmites australis Lineages

The three dominant *P. australis* lineages of the northern Gulf of Mexico coast exhibited similarly reduced survival in response to increased flooding. However, different haplotypes displayed divergent mortality trends; the Delta haplotype had the highest survivorship after two years in the saline site, yet the lowest survival in the fresh site (Figure 6). While Delta is the dominant *P. australis* haplotype across the Gulf Coast and in the natural marshes of the fresh MRD site, it may have suffered more in the marsh organs at the fresh site due to lack of clonal integration from a preexisting dense rhizome network [40]. Delta *P. australis* was the only haplotype with any surviving individuals in the three low-elevation treatments of Rockefeller Wildlife Refuge after two growing seasons, suggesting that the Delta haplotype is the most robust to coincident flooding and salt stress and that it is the haplotype most amenable to restoration following dieback (also supported by [41]). This is promising since Delta is the most widespread *P. australis* lineage in the MRD [16,42]. Additionally, the Gulf haplotype exhibited a distinctly less negative response to flooding in its aboveground biomass compared to other haplotypes (Figure 2).

### 3.2. Response to Abiotic Stress

Flood stress response was more extreme in the saline site (Figure 6), similar to findings for a more salt-tolerant marsh species, *Spartina patens* [22], as well as fresh marsh species [39]. Salinity concentrations of 20 ppt can induce stress in all *P. australis* lineages and are deadly to Gulf *P. australis*, while concentrations of 30 ppt result in mortality for Delta and EU *P. australis* [18]. Our marsh organ study demonstrates how multiple abiotic stressors—salinity and flooding associated with sea-level rise—work together to contribute to marsh plant mortality. Sea-level rise-associated flooding often induces stress in wetland plants by reducing available oxygen in the rhizosphere and lowering the porewater redox potential [43], which in turn can influence anaerobic respiring bacteria to produce phytotoxins such as sulfides and volatile fatty acids by metabolizing dead plant matter [44]. Previous researchers found that flooding negatively impacts *P. australis* growth vigor [45], while our results show distinct negative effects of flooding on *P. australis* survival, yet not necessarily on biomass accumulation. *P. australis’* general linear decline in survival with increasing flooding contrasts with the parabolic response of biomass accumulation in another fresh–brackish marsh species, *Schoenoplectus americanus* (Pers.) Volk. ex Schinz & R. Keller [24]. Overall, the poor survivorship and biomass production in the saline site can be attributed to the compounding negative effects of anaerobic conditions, high sulfides [46], and high salinity [18,47]. This is especially evident in the low-elevation treatments simulating high levels of sea-level rise in the saline site, where we observed very low survival and biomass production (Figure 6 and Figure 7) as all these stressors were simultaneously exerting their influence on the plants (Figure 3b and Figure 4).

Acetic acid porewater concentration increased with greater flooding across both sites and higher acetic acid concentration was associated with a greater probability of mortality (Figure 3), suggesting that acetic acid buildup is one of the mechanisms by which increased flooding contributes to *P. australis* dieback, even under low-salinity conditions. Volatile fatty acids such as acetic acid are potent phytotoxins that accumulate in marsh soils due to anaerobic fermentation of dead plant material [48] and have been linked to *Phragmites australis* dieback in central European wetlands [49]. Although this trend was detected in our study, volatile fatty acids will likely accumulate more in fresh marshes due to faster decomposition of cellulose compared to marshes with strong seawater influence [50]. *P. australis* in the marsh organs had much higher porewater acetic acid concentrations than those sourced from healthy and damaged *P. australis* stands in Europe (6.3 and 24.3 ppm, respectively; Ref. [49]), and those used to induce dieback conditions in a laboratory (100 ppm; Ref. [35]). Other volatile fatty acids, such as heptanoic and butyric acid, were not significant predictors of *P. australis* mortality in the marsh organs; thus, we agree with Kovacs et al. (1989), who posit that acetic acid is the most harmful volatile fatty acid for *Phragmites*. However, larger molecular weight VFAs are considered more harmful to plants in general [51]. Wetland soils high in calcium may help to protect against acetic acid toxicity, as calcium carbonate can neutralize harmful acetic acid, and calcium ions can chelate the organic acids [52]. Porewater nutrient analysis showed calcium levels were generally lower in the most flooded treatments (Appendix A), indicating potential depletion from chelating the acetic acid. This suggests that in situ calcium levels in *Phragmites* marshes are not high enough to substantially ameliorate the negative effects of acetic acid buildup.

We found strong evidence that porewater sulfides contribute significantly to plant stress in both low-salinity and high-salinity conditions (Figure 8), and are especially prevalent in high-salinity *Phragmites* rhizospheres (Figure 4). Hydrogen sulfide is another strong phytotoxin prominently associated with wetland plant stress and dieback [35,53]. Sulfide toxins are usually associated with saltwater intrusion into coastal marshes [54], and our marsh organ experiment showed dramatically higher sulfide concentrations in the saline site compared to the fresh site, likely a contributing factor to the vastly reduced survival in the saline site. Belowground biomass decreased with an increase in porewater sulfide concentration in the freshwater site, and aboveground biomass similarly decreased in Rockefeller (Figure 8), demonstrating that sulfide can negatively impact plants in coastal environments, even with lower levels of salt-water intrusion. The sulfide concentrations detected in marsh organs were much higher than those found in natural *Phragmites* stands in the Mississippi River Delta (~145 μmol/L [55]). Aeration of the rhizosphere through exposure of the sediment to oxygen can relieve sulfide toxicity by oxidizing the compound to non-toxic sulfate, and the marsh organ pipes may have prevented rhizosphere aeration, leading to the observed high sulfide levels. Regardless of inflated sulfide levels in the marsh organs, this illustrates how sea-level rise will continue to harm wetland plants—through increased periods of inundation with less chance for sulfide oxidation and neutralization. Additionally, soils high in iron may mitigate sulfide toxicity by iron binding to the sulfur and precipitating iron sulfide [56,57,58]. Iron levels in the most flooded treatment of Rockefeller were extremely low, indicating that all the available iron may have already bound to the accumulating sulfide in the high flooding, high-salinity treatment (Appendix A).

Sulfide reached a clear concentration threshold past which none of the experimental plants survived (Evident in Figure 8, around 3800 μmol/L of sulfide), while most plants did not survive concentrations greater than 1000 μmol/L. Thus, while our statistical tests did not reveal a significant impact of sulfide concentration on survival probability, it does appear that the two are related. Previous research shows that in natural marsh conditions, porewater sulfide concentrations greater than 400 μmol/L are associated with reduced growth resulting in shorter *Phragmites* shoots [59].

Sulfide concentration increased substantially from the first to the second growing season, coinciding with a decrease in survival. During the first growing season, sulfide concentration was relatively low and did not differ significantly between the high-salinity Rockefeller site and the low-salinity MRD site (Figure 4). By year two, sulfide concentrations were much higher in the saline site, coincident with greatly reduced survivorship and biomass (Figure 8). Sulfides are produced by rhizosphere bacteria under anaerobic conditions [56], and such production may be accelerated by the death and decay of the plant’s roots or through eutrophication and organic loading [30]. We posit that *Phragmites* dieback can operate as a feedback loop, such that increased stress leads to root and rhizome death, which then act as a substrate for bacteria to anaerobically produce toxins such as sulfides and volatile fatty acids that continue to harm the plant [60]. If sea-level rise leads to coastal wetlands experiencing longer periods of continuous inundation, the resultant anaerobic soil environments and associated sulfide accumulation will be severely detrimental to marsh plants, possibly leading to dieback.

### 3.3. Management Implications

In demonstrating abiotic stressors of *Phragmites* related to sea-level rise, our study also provides clues for managing invasive *Phragmites* in areas where the plant is unwanted. We highlight here the role that inundation plays in *Phragmites* stress and mortality, which matches with evidence suggesting that mowing and flooding are viable for controlling the spread of the plant [45,61]. *Phragmites* occurring at the upland fringes of salt marshes will be more vulnerable to increased flooding with SLR than those occurring in lower salinity marshes. We also suggest that stress from future sea-level rise could inhibit *Phragmites’* growth in regions where it is considered invasive, instead favoring the growth of more salinity- and sulfide-tolerant species such as *Spartina alterniflora* [62]. Combined with our findings detailing the stress-inducing effects of sulfide and acetic acid, this suggests the possibility of managing *Phragmites* invasion using more eco-friendly methods as opposed to herbicides, which can have deleterious non-target toxic effects in wetlands [63,64]. On the other hand, if not replaced by another species of marsh grass, sea-level rise-associated extirpation of *P. australis* will also deprive coastal wetlands of important ecosystem services such as a high accretion rate, which could lead to marsh loss [28].

## 4. Materials and Methods

### 4.1. Study Locations

Two *P. australis*-dominated marsh sites along the Gulf Coast of Louisiana that differ in seawater exposure—Rockefeller Wildlife Refuge (RWR) located in the Mermentau basin along the southwest coast of Louisiana representing the brackish site, and the Delta National Wildlife Refuge located in the lower Mississippi River Delta (also known as Birdsfoot delta, hereafter MRD) along the southeast coast of Louisiana—were selected as study locations. RWR has no freshwater riverine influence, and recent storm events such as Hurricane Laura (2020) have increased the salinity in the area; during our study, the salinity at RWR averaged 17.6 ± 0.2 (SE) ppt. On the other hand, the MRD consists of mostly fresh marshes due to the abundance of freshwater delivered by the Mississippi River. Due to relatively predictable seasonal river discharge, coastal set-up, and frequency of tropical cyclones, salinity in the MRD tends to be lower in the spring and higher in the late summer through fall. Over the course of this study, the salinity at the MRD site averaged 1.6 ± 0.04 ppt. Both sites are dominated by the Delta haplotype of *Phragmites australis.*

At least four morphologically distinct *P. australis* lineages coexist in marshes along the Louisiana Gulf Coast, three of which are the focus of our study. Gulf haplotype *P. australis* is seldom found in low-lying coastal marshes, yet is widespread along roadside ditches and canal banks inland. Delta is the most widespread *Phragmites* haplotype in Louisiana and is abundant in fresh to brackish marshes. European *Phragmites* are relatively rare in the MRD, only found in the MRD at slightly higher elevations than the Delta haplotype [55].

### 4.2. Plant Collection

We collected 50 cuttings of Gulf and Delta varieties from the sea-water-exposed sites (RWR) and 50 cuttings of Gulf, Delta, and EU varieties from the low-salinity site (MRD) between February and April 2021, for a total of 250 plants. We excavated stem base and rhizome clumps, clipped the live and dead stems to ~1 m, and then divided them into approximately equal sections with ~5” of live intact rhizome material to form similar-sized replicates. We planted each cutting into a 6” diameter plastic pot using soil collected from a delta haplotype *P. australis* stand near Venice, LA. Voucher specimens from each *Phragmites australis* haplotype used in this experiment are deposited in the Shirley C. Tucker Herbarium at Louisiana State University as accession numbers LSU00218181 (EU), LSU00218183 (Gulf), and LSU00218185 (Delta).

### 4.3. Experimental Design and Marsh Organs

Five marsh organs consisting of five rows of five 15 cm diameter PVC pipes were placed in each site. To calculate the desired height of each PVC pipe for each marsh organ, we used site-specific hydrologic data close to the study locations for the 2014–2019 growing seasons (1 March–31 October) using the Coastwide Reference Monitoring System (CRMS) sites 4448 and 0615 for the MRD and RWR, respectively. Information on the CRMS stations can be found at https://www.lacoast.gov/crms/FAQ.aspx (accessed on 20 July 2023). We sorted the water levels into quantiles of 0.95, 0.85, 0.60, 0.30, and 0.10, representing elevations inundated 5%, 15%, 40%, 70%, and 90% of the time, respectively. PVC pipes were cut to respective lengths for each inundation level including extra length for anchoring into the sediment. Then, we bolted pipes together in rows of equal height and columns of increasing height.

To install the marsh organs in the field, we placed the tallest pipes (lowest inundation level) oriented north to avoid shading the plants in the smaller pipes [22]. In the field at both RWR and MRD, we placed the marsh organs into the sediment and used a real-time kinematic GPS to determine the specific elevations to place the pipes (Leica CS30, Leica Geosystems, Heerbrugg, Switzerland). Pipes were then filled with root-free sediment collected from adjacent delta haplotype *P. australis* patches. Thus, marsh organs had local soil from respective sites. Each pipe was planted with a *P. australis* plant growing in a 6″ diameter pot with each row receiving one of each haplotype/source location combination (i.e., Delta-type Phragmites from the MRD), planted in random order. We planted experimental plants into the marsh organs at Rockefeller on 14 May 2021 and at the MRD on 3 June 2021.

### 4.4. Measurements

#### 4.4.1. Flooding Dynamics

To determine the percent time flooded of each elevation treatment in the marsh organs over the study period, we used an RTK GPS (Leica CS30 and Viva GS14, Leica Geosystems, St. Gallen, Switzerland) to determine the elevation relative to NAVD88. Daily mean water levels at each CRMS station nearest to the marsh organs were used to calculate the percentage of time inundated and mean water depth for each row.

#### 4.4.2. Porewater Chemistry and Redox Potential

In total, 5 porewater samples were collected (1 from each elevation) 20 cm below the soil surface from each of the five organs, for 25 porewater samples at each site. At the two highest elevations in the MRD organs, the soil was too dry, so porewater samples were collected from a 50 cm depth. Porewater was stored on ice until it was placed in the freezer (nutrient and VFA samples) or fridge (sulfide samples). We collected porewater during the survivorship assessment in September–October 2021 (~4 months after the start of the experiment) and prior to the biomass harvesting in August–October 2022 (~1.25 years after the start of the study). Because we were only able to collect porewater from a subset of marsh organ wells, we were not able to detect differences in porewater phytotoxins among haplotypes. Redox potential was measured at the end of the second growing season prior to harvesting the plants at a 20 cm soil depth using a field redox probe. Field electrodes consisted of a Pt electrode attached to the positive end of a high resistance meter and a reference Ag-AgCl electrode attached to the negative end of the meter [65]. The electrodes were allowed to equilibrate for several minutes before redox potential was recorded.

We analyzed porewater nutrient content (phosphate-P, nitrate + nitrite-N, ammonium-N, Ca, Fe, S, K, Mg, Mn, Na, Si, and SO_4_) using a SEAL AA-500 (SEAL Analytical, Mequon, WI, USA) autoanalyzer and metal content using Varian Vista MPX ICP-OES (Varian Inc., Palo Alto, CA, USA). To measure concentrations of volatile organic acids, we ran 8 mL of filtered porewater on a microFAST gas chromatograph (ASI Inc., Baton Rouge, LA, USA [66]), and compared it to a 60 ppm standard solution with acetic, propionic, isobutyric, butyric, isovaleric, valeric, isocaproic, caproic, and heptanoic acid. To measure sulfide concentrations of all sulfide species combined, we preserved sulfide in porewater by adding 1 part zinc acetate solution to 2 parts porewater, then used spectrophotometry at 670 nm to determine sulfide concentration with the methylene blue method [67,68,69].

#### 4.4.3. Survivorship and Productivity

Survivorship was measured in year one approximately 4 months after the start of the study (20 September in Rockefeller and 1 October 2021 in MRD). After two growing seasons, we harvested all aboveground biomass and the top 50 cm of belowground biomass from the MRD marsh organs on 21 September and 7 October and from the Rockefeller marsh organs on 4–5 August 2022. Following the harvest, sediment was rinsed from the belowground biomass, and above- and belowground biomass was separated into live and dead categories and dried in a drying oven at 60 °C to a constant weight.

### 4.5. Statistical Analysis

To test how the main effects of increased inundation time, lineage, and site (along with all interactions) influenced *Phragmites'* survival in the marsh organs, we ran separate logistic regressions for each growing season using R statistical software (version 3.6.3 used for all analyses; Ref. [70]; Figure 2). To test how flood duration influenced plant performance, we separately calculated the percent time flooded from the time of the initial planting to the first data collection during year one, and from the initial planting time through the final harvest for year two.

We then used logistic regressions to separately test in each site how porewater concentrations of sulfide and each volatile fatty acid predict the survivorship probability during each growing season. We only tested the effects of VFAs on survival in the first growing season due to failure to detect VFAs in porewater collected during the second growing season. Finally, we performed a logistic regression to test if redox potential predicted the probability of survival by the end of the second growing season.

To test how the main effects influenced biomass accumulation in marsh organs, we first used linear models followed by ANOVA (type III Wald chi-square tests) to regress the percent time flooded, lineage, and site (and all interactions) on above- and belowground biomass separately. We then ran separate tests in both sites to regress percent time flooding, sulfide concentration, and lineage on above- and belowground live biomass. All tests included marsh organ as a random effect to account for experimental design. If the interaction was not significant, we removed it from the formula and re-ran the model. If nothing in the model was significant, we tested the effects of flooding and sulfide on biomass using separate models. We used a +1 log transformation on the biomass, and a standard log transformation on the sulfide concentration to scale the variables and meet the assumptions of ANOVA.

Additionally, we conducted ANOVAs to test if porewater redox potential is a significant predictor of live aboveground or belowground biomass accumulation, as well as sulfide and acetic acid concentrations. We also tested the relationship between the percent time flooding and redox potential. All data from both sites were included in the test, and we controlled for the site and marsh organ by including the site as a covariate for the fixed effect in the model, and the marsh organ as a random effect in the model. If the model yielded no interaction between the site and the specified fixed effect, then we removed the interaction term and re-ran the model. We used separate paired t-tests for each site to check for differences in sulfide concentrations across years, as we collected sulfide samples from the same marsh organ wells each year.

To test how nutrients differed across treatments during the first growing season, we performed a type III ANOVA testing how inundation time, site, and the interaction between the two influenced porewater nutrient concentrations of Ca, Fe, K, Mg, Mn, Na, S, Si, and sulfate (Appendix A). For Ca, Si, and SO_4_, we log transformed the concentration to adhere to the assumptions of ANOVA. For Fe, we used a + 1 and then log transformation to successfully transform zeros in the dataset. For the porewater samples collected at the end of both the first and second growing seasons, we compared phosphate, nitrate, and ammonia concentrations across inundation treatments and sites (Appendix A).

## 5. Conclusions

Here, we demonstrate clear evidence that increased flood duration and associated abiotic stressors contribute to reducing *Phragmites australis* biomass accumulation and survival. We also show the *P. australis* stress response to flooding was exacerbated in high salinity compared to low-salinity conditions. Furthermore, our results detail how the *P. australis* haplotypes respond differently to sea-level rise-associated abiotic stressors, in particular, the delta haplotype is the most robust to simultaneous high salinity and extended inundation conditions. We suggest that marsh dieback will accelerate under rising sea levels and tropical storm-induced saltwater intrusion, both of which are projected to increase with continued climate change. These findings will be valuable for managing wetland loss in the Mississippi River Birdsfoot Delta and other regions threatened by sea-level rise and will help formulate strategies to mitigate the negative effects of *Phragmites* invasion in other parts of the world.

## Figures and Tables

**Figure 1 plants-13-00906-f001:**
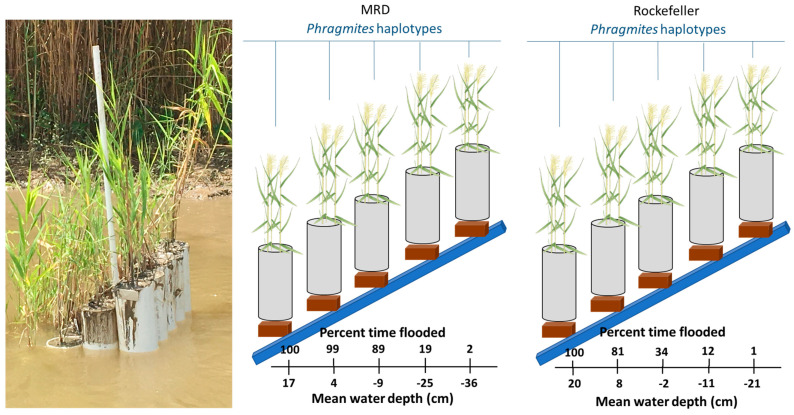
Marsh organ percent time flooded, and water depth across both sites. Differences in percent time flooded between sites were due to the influence of river (MRD) versus coastal bay with limited exchange (Rockefeller) hydrodynamics.

**Figure 2 plants-13-00906-f002:**
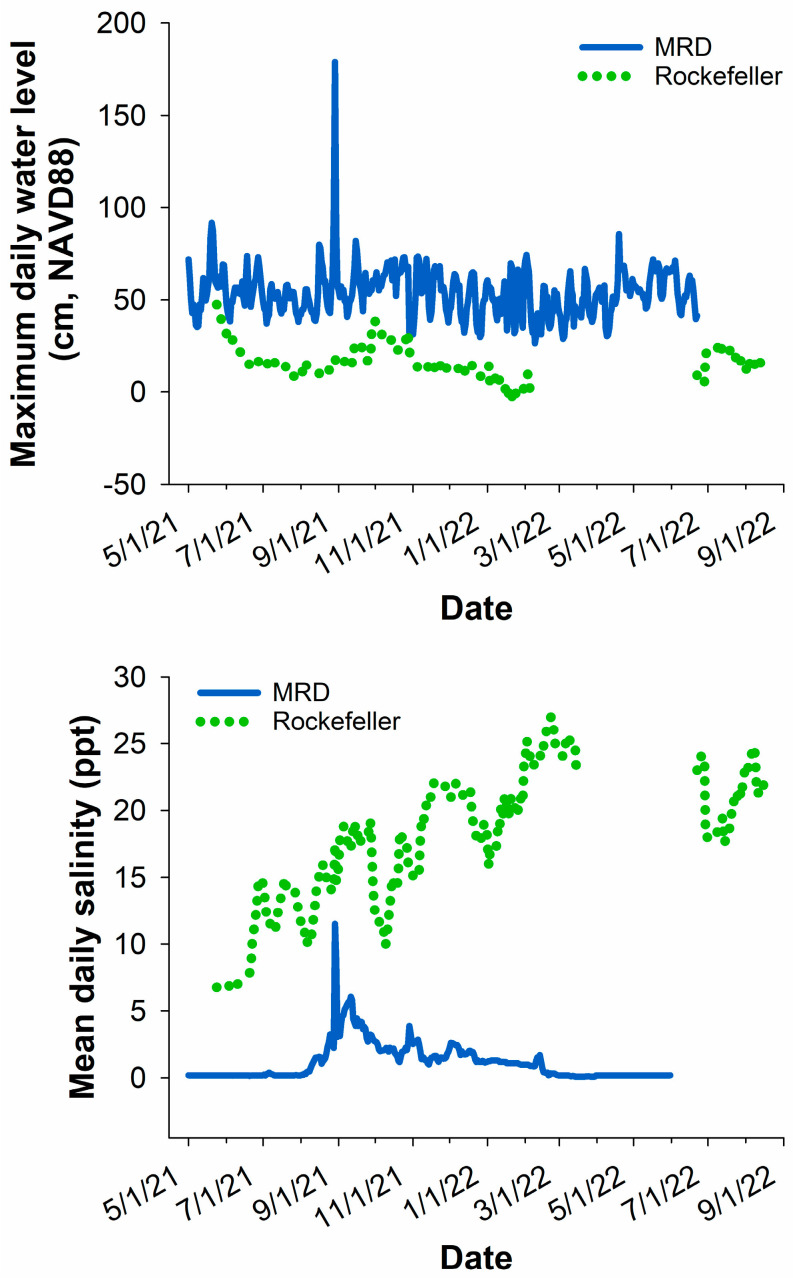
Flooding and salinity levels in each site during the marsh organ study period in the Mississippi River Delta (MRD) and Rockefeller Wildlife Refuge. Some data are missing at Rockefeller due to errors with the Coastwide Reference Monitoring System data loggers.

**Figure 3 plants-13-00906-f003:**
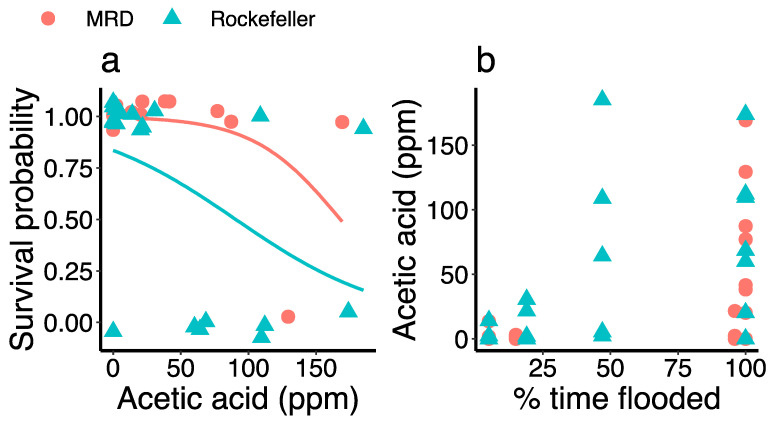
(**a**) Logistic regression results showing how acetic acid concentration predicts the probability of survival on *Phragmites australis* in the marsh organs from both the Mississippi River Delta (red circles) and Rockefeller Wildlife Refuge (blue triangles); (**b**) scatterplot displaying the relationship between percent time flooded and porewater acetic acid concentration in the first growing season.

**Figure 4 plants-13-00906-f004:**
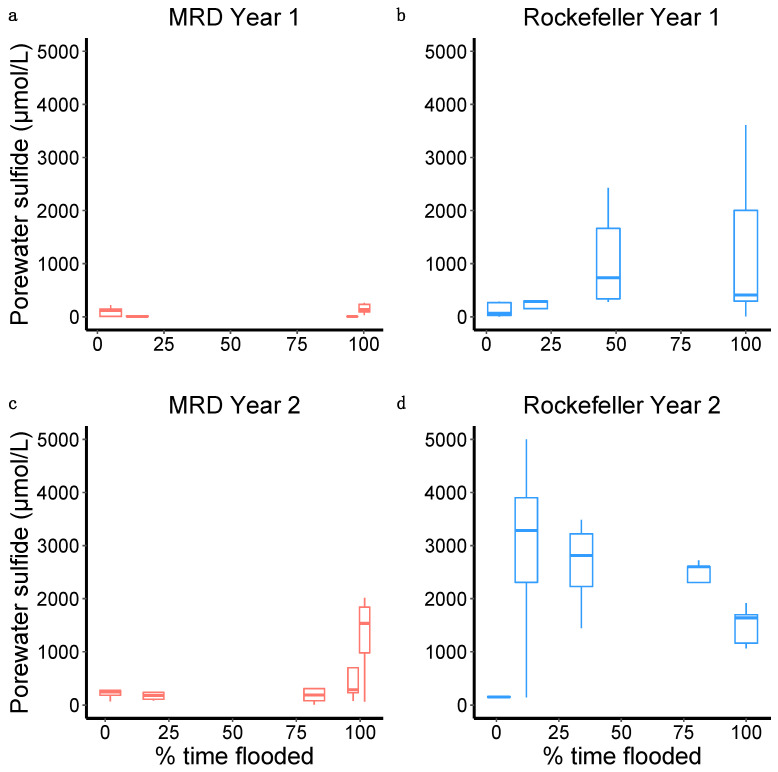
Boxplots displaying the first growing season (**a**,**b**) and second growing season (**c**,**d**) relationship between percent time flooded and porewater sulfide concentration in the experimental marsh organs placed in the MRD and in Rockefeller Wildlife Refuge.

**Figure 5 plants-13-00906-f005:**
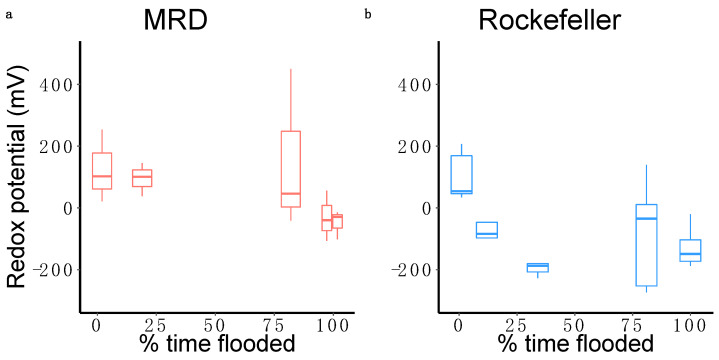
Boxplots displaying the relationship between percent time flooding and redox potential recorded at the end of the second growing season in the Mississippi River Delta (**a**) and Rockefeller Wildlife Refuge (**b**).

**Figure 6 plants-13-00906-f006:**
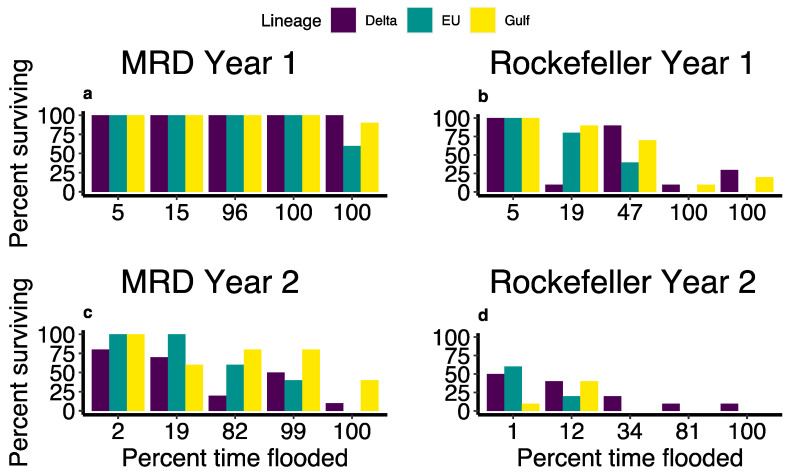
*Phragmites* transplant survivorship following the first (**a**,**b**) and second (**c**,**d**) growing season in the marsh organs at the Mississippi River Delta (MRD; **a**,**c**), and Rockefeller Wildlife Refuge (**b**,**d**).

**Figure 7 plants-13-00906-f007:**
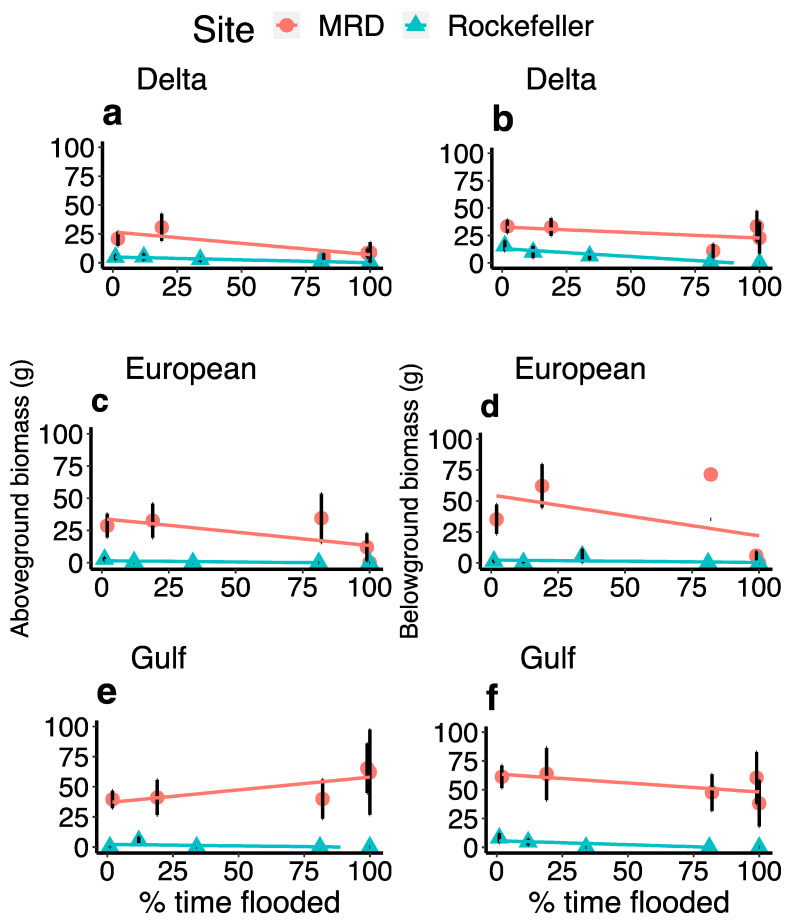
Plots displaying the relationships between percent time flooded and either aboveground or belowground biomass. Plots (**a**,**b**) show results from Delta *Phragmites*, plots (**c**,**d**) show European *Phragmites*, and (**e**,**f**) display Gulf Phragmites. The Mississippi River Delta data is represented by red circles, while Rockefeller Wildlife Refuge is represented by blue triangles.

**Figure 8 plants-13-00906-f008:**
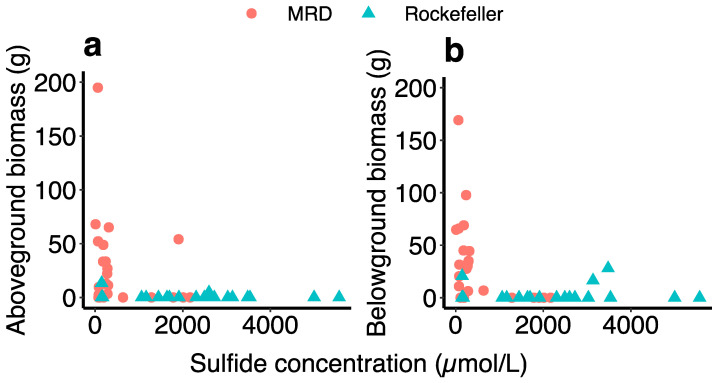
Scatterplots displaying the relationship between sulfide concentration and biomass (g), separated into above (**a**) and belowground biomass (**b**).

## Data Availability

Data will be uploaded to the Zenodo digital repository at https://doi.org/10.5281/zenodo.10822162.

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
