# Peer review of "Salt Water Exposure Exacerbates the Negative Response of Phragmites australis Haplotypes to Sea-Level Rise"

_plants, 2024, doi:10.3390/plants13060906_

Round 1
Reviewer 1 Report
Comments and Suggestions for Authors
The authors observed the responses of Phragmites australis to seawater inundation and exposure by incubation experiment. Considering various haplotypes subjected to both inundation and salinity gradients and the common experiments obtained abundant interesting results, this manuscript deserves publication after revision.
Specific comments:
-Abstract: In addition to qualitative descriptions, more quantitative data should be added.
L18-21: How long did the incubation last?
-Introduction:
L91-93: The 3rd hypothesis is unclear. Why not test the differences between native vs invasive lineages?
-Results: Since ANOVA was mentioned in the statistical analysis, but there are almost no significant differences reflected in the relative graphs?
Figure 1: The salinity difference could be added between the two sites.
Figure 2: The data missed should be explained in the caption and text. Salinity unit should be added too.
-Discussion: More results supported by the figures should be cited in this section, in addition to references stacking.
-Materials and Methods:
L383-384: More detailed description of the soil should be added. e.g., chemical compositions. Which haplotype's origin is closer to the environmental background of this stand near Venice, LA? If the soil has a preference, it is difficult to avoid the home effect.
-Conclusions: Please directly answer the there hypotheses in Introduction.
Comments on the Quality of English LanguageNo.
Reviewer 2 Report
Comments and Suggestions for Authors
The article Salt Water Exposure Exacerbates Negative Response of Phragmites australis Haplotypes to Sea Level Rise written by Austin Lyn and Tracy Quirk provides interesting dates.
The research conducted on different haplotypes of Phragmites australis holds significant value in shedding light on the survival mechanisms of these plant variations in the face of climate change. Particularly, the study delves into their ability to thrive amidst challenging conditions such as flooding and rising sea levels, offering crucial insights for future conservation efforts. Understanding which haplotypes exhibit resilience and adaptability to salinity conditions is pivotal for effective restoration initiatives.
The comprehensive analysis of graphs and figures in the study enhances the clarity and accessibility of the findings.
However, In Figure 2, it would be beneficial to include the meaning of MRD (Mississippi River Delta) for improved comprehension: ensuring that all graphs are self-explanatory can enhance the overall accessibility of the research data.
As a suggestion for improvement, replace "saw" with "observed" in Line 268. This change would align with scientific terminology and enhance the accuracy of the report.
In some of the species named in the text the corresponding authors do not appear. This should be corrected
Finally, this research not only contributes to our understanding of plant resilience in changing environments but also provides valuable guidance for conservation and restoration practices in the face of climate challenges.
